# Emotional Self-Regulation in Primary Education: A Heart Rate-Variability Biofeedback Intervention Programme

**DOI:** 10.3390/ijerph19095475

**Published:** 2022-04-30

**Authors:** Aitor Aritzeta, Ainara Aranberri-Ruiz, Goretti Soroa, Rosa Mindeguia, Amaiur Olarza

**Affiliations:** 1Department of Basic Psychological Process and Development, University of the Basque Country (UPV/EHU), 20018 San Sebastian, Spain; ainara.aranberri@ehu.eus (A.A.-R.); rosa.mindeguia@ehu.eus (R.M.); amaiur.olarza@ehu.eus (A.O.); 2Department of Clinical and Health Psychology and Research Methodology, University of the Basque Country (UPV/EHU), 20018 San Sebastian, Spain; goretti.soroa@ehu.eus

**Keywords:** biofeedback intervention, heart rate variability, children, polyvagal theory, HeartMath emWave software

## Abstract

This study investigated the benefits of using a biofeedback intervention programme to train children in controlling their heart rate variability (HRV) through slow-paced breathing in real time. HRV biofeedback interventions focused on showing subjects to breathe such that their HRV numbers rise, improving their self-regulation. The HRV biofeedback intervention, focused on breathing, was conducted with primary education students aged between 7 and 11 years. The programme consisted of five biofeedback sessions, where students were taught to breathe six long and slow pairs of breaths per minute, to increase their HRV. After participation in the programme, students, regardless of gender, increased their HRV in a statistically significant fashion with a large effect, but this effect was not the same for all ages. HRV biofeedback interventions are rarely applied in schools and given the effectiveness of the intervention to improve HRV in children, the applied implications of our results in educational settings are discussed, especially taking into account the children’s ages.

## 1. Introduction

In childhood, emotions are experienced with great intensity and with low capacity for emotional regulation [1] which may have a negative impact on psychosocial skills, school performance and well-being [2,3]. During primary education, the capacity for emotional self-regulation is still under development which offers an opportunity to implement interventions with the goal of improving emotional self-regulation abilities [4]. Specifically, interventions based on biofeedback of the HRV and focused on breathing have been shown to be effective in improving emotional self-regulation capacity in children [5].

Biofeedback is a broadly used method to train people to voluntarily control certain physiological functions, such as breathing, which consists of providing users with instantaneous information on variations that occur in their own physiological activity [6]. Using heart rate variability-based biofeedback programmes, by means of breathing practise, the subjects learn to breath such that heart rate variability (HRV) increases [7,8]. In this regard, it was observed that adequate HRV biofeedback teaches people to breathe at a frequency of approximately six breaths per minute [9,10].

As far as age is concerned, it has been observed that in children the degree of HRV and self-regulation capacity are related. Different studies have shown that greater HRV reflects better psychosocial adjustment in the childhood population [11,12]. Children with behaviour problems display lower HRV [13,14], however, we do not have enough evidence about the differential effectiveness that, regarding age, a biofeedback training programme based in breath pacing, may have on children from 7 to 11 years old in Primary Education.

In order to offer a comprehensive theory on children’s capacity for self-regulation through biofeedback programmes, the Polyvagal Theory should be mentioned. This theory addresses the interrelation between the vagus nerve and the emotional experience allowing us to associate self-regulation abilities with values of HRV. The vagus nerve is the tenth cranial nerve and is the main nerve in the parasympathetic division determining emotional self-regulation [15]. Primary education children completely possess the vagus nerve system [16,17] and the ventral-vagus branch and the dorsal-vagus branch of the vagus nerve, provide an inhibitory entry to the heart through the parasympathetic nervous system (reducing heart rate) and influence HRV outcomes.

The vagus nerve system is hierarchically structured into three sequential functional sub-systems: (a) The ventral vagal complex or myelinated vagus, (b) The sympathoadrenal system, and (c) The dorsal vagal complex or non-myelinated vagus [18,19]. When the environment is perceived to be safe, the ventral vagal complex is activated, leading to an increased influence of the myelinated vagal channels that slow down heartbeat frequency, increase HRV, and inhibit fight-or-flight mechanisms of the sympathetic nervous system. This physiological state makes it possible for the prefrontal cortex’s structures to work properly, which are in charge of attention and self-regulation, both of which are differentially developed for a child of 7 years old and one of 11 years old. However, and despite primary education children completely possessing the vagus nerve system, we still do not have enough evidence to know how effective an HRV and breathing based biofeedback programme can be dependent on the age of the children

The aforementioned HRV can be assessed with several analytical foci, although the most commonly used are frequency domain analysis (power spectral density) and time domain analysis [20,21]. As it has consistently been shown in the literature, HRV is considered a measurement of physiological emotional self-regulation [22,23,24,25,26] and a biomarker for psychopathology [27]. High HRV also has been related with lower levels of frustration and higher performance levels [28] and positive psychological adjustment in children, adolescents, and the adult population [27,29], including, for example, empathetic responses to other people in danger [30,31], social competence [32], and positive social interactions amongst equals [33].

Low HRV is the result of supressing the vagal brake. Suppression of the vagal brake entails disappearance of the parasympathetic influence wielded by the ventral vagus, and increased cardiovascular pace, which reduces HRV. The subjective emotional experience under this neurophysiological state is a variable degree of tension, malaise, and anxiety [34,35]. In terms of psychological regulation, reduced HRV has been associated with reduced capacity for self-regulation and cognitive functions that are primarily located in the prefrontal cortex [26,36,37,38].

Through breath control, healthy people can increase their own HRV by modifying respiratory frequency, through slower and steadier breathing [39,40] by means of the vagus nerve’s influence on the sinoatrial node of the heart, slowing the heart rate, and increasing HRV [36,41,42]. The high HRV generated through voluntary breathing is a suitable emotional self-regulation measure [43,44].

HRV biofeedback can be conducted by placing a device on a person that connects to a computer and provides real-time information on their HRV. By observing the impact that breathing has on HRV in real time, one learns to breathe (through trial and error and feedback), and to improve their emotional self-regulation capacity [7]. In this sense, scientific literature has shown that biofeedback interventions focused on breathing, increase HRV both in the adult population [45,46,47,48,49,50] and in the childhood population [50,51]. For example, Van der Zwan et al. [52] conducted an HRV biofeedback intervention focused on breathing for 5 weeks, analysing the programme’s effect on the well-being of pregnant and non-pregnant women. After the intervention, both groups had reduced anxiety and increased well-being. In order to better tackle stress during exams, Deschodt-Arsac et al. [53] conducted a 5-week intervention on HRV biofeedback focused on breathing, with university athletes. After the intervention, the experimental group significantly improved HRV. This improvement was maintained for 12 weeks. Poskotinova et al. [54] conducted an HRV biofeedback intervention in one single session with 20 Secondary School students (15 and 16 years old), in order to increase the participants’ HRV as a protective resource against internet addiction disorders. After the intervention, participants’ HRV was significantly increased, and correlation analyses showed a significant negative correlation between increased HRV and scores obtained in measuring internet addiction. Mei et al. [9] conducted an intervention with 32 adolescents between 13 and 18 years of age, also in one single HRV biofeedback session. After the intervention, a significant increase in HRV was observed, and more significant than another group of students who received alternative treatment in an autogenous training session.

In this realm, other interventions in HRV biofeedback were also proven effective. For example, in terms of the university population, HRV biofeedback interventions focused on breathing, improved participants’ HRV [45,49], and in terms of the school-aged population, students participating in HRV biofeedback interventions also improved HRV [47,50,51,55].

Therefore, literature consistently shows biofeedback interventions have a positive impact on participants HRV, especially in adolescents and adults. Less evidence is gathered on the impact of these programs in children from 7 to 11 years old. Therefore, the main goal of this research was to examine the effectiveness of a biofeedback breath pacing training programme on HRV in children at primary school, taking into account the age of the participants.

Based on previous results, we expect that; compared to a control group, the values of the HRV of the experimental group will be higher than the control group values after the intervention.

Regarding the above-mentioned neuro-anatomical development processes, we will explore if the lower neuro-anatomical maturity progress at lower ages might also affect the effectiveness of the intervention programme, so as to observe better HRV results as the age of the participants increases.

Finally, in terms of gender, we did not find similar research conducted on school-aged populations to examine differences based on gender, which is possibly explained by limited sample sizes. In other age ranges, Hill et al. [56] examined 172 studies mainly focused on the adult population. The results showed that, in comparison with men, women displayed greater amplitude in HRV. Brunetto et al. [57] conducted a study with 41 adolescents (20 boys and 21 girls), aged 12–17 years, where no statistically significant gender differences were found in HRV scores. Finally, in the study conducted by Aziz et al. [58], upon comparing HRV parameters between 33 girls and 37 boys, they observed that only boys had lower scores than girls in only three out of 11 HRV parameters (SDNN, LF, and SD2). Therefore, we also aim to investigate if there were gender differences in HRV values before and after the intervention programme.

## 2. Materials and Methods

### 2.1. Sample

The experimental and control groups were divided into the three cycles: 29.2% belonged to the first cycle (2nd grade, M = 7.58 years and SD = 0.38), 57.7% to the second cycle (3rd and 4th grades, M = 8.93 years old and SD = 0.85), and the remaining 13.1% were third-cycle students (5th and 6th grades, M = 11.04 years old and SD = 0.91).

Participants were studying in primary education in two different public schools using the Amara Berri Education System which seeks to educate students using everyday problems and organizing their everyday learning process in thematic spaces (Murumendi and Larrea public schools). The experimental and control groups were divided in the three cycles: 29.2% belonged to the first cycle (2nd grade, 7 years old), 57.7% to the second cycle (3rd and 4th grades, 8–9 years old), and the remaining 13.1% were third-cycle students (5th and 6th grades, 10–11 years old).

Student participation was voluntary and consented to by the school board, parents, and guardians. Classrooms were randomly assigned to experimental and control groups, however, all children, except those with a clinical diagnosis (i.e., hyperactivity, depression or anxiety disorders) belonging to those classrooms participated in the study. So, children were not randomly assigned to the treatment/non-treatment conditions.

The study had a favourable report from the ethics committee for research with humans, their samples, and their data (M10-2020-318) from the University of the Basque Country/Euskal Herriko Unibertsitatea (UPV/EHU). Ethical aspects required for research with humans were scrupulously followed (informed consent, right to information, personal data protection, confidentiality guarantees, non-discrimination, no cost, and possibility to leave the study during any of its phases).

### 2.2. Design

Given the inherent characteristics of the natural groups (primary education classrooms) and the natural context (intervention takes place at the school), a quasi-experimental intervention was designed with a control group. Five HRV scores were taken from all experimental participants. The first measurement of session one (S1) is the pre-treatment measurement and informs about the baseline HRV numbers. The fifth measurement of session five (S5) is the post-treatment measurement. For the control group only two measures were taken in session one (S1) and session five (S5) and at the same period of time as the experimental group.

### 2.3. Instruments and Materials

As follows, a description of the instruments and materials used are described.

Recording sheet (ad hoc): A document was specifically prepared to record the data obtained from each student in each session throughout the intervention. It contains the HRV scores from the five sessions.

HRV: HeartMath EmWave software was selected for this study [59], whose efficacy has been proven in different studies [5,50,55]. This is a non-invasive instrument that measures HRV in real time through a sensor placed on the earlobe. Using a mathematical algorithm, the EmWave software transforms the collected data (e.g., RMSSD, SDNN...) into low, medium, and high coherence ratios. These low, medium and high coherence ratios are directly related to HRV amplitude rates. Thus, a low coherence level matches low HRV, a medium coherence level matches medium HRV, and a high coherence level matches high HRV.

After each session, the application shows the HRV scores by distributing 100 points in three levels (1) A low coherence level, or low HRV, shown in red; (2) An intermediate coherence level, or intermediate HRV, shown in blue; and (3) A high coherence level, or high HRV, shown in green. These scores are set forth on the record sheet and used for later statistical analyses.

Two EmWave software applications were used [59]: *Coherence Coach* and *Balloon Game*. By applying Coherence Coach, children learn to breathe following the rhythm of a moving ball on the computer screen, learning to breathe six breaths per minute in the following fashion: when the ball goes upward, participants must breathe in through the nose, and when the ball goes down, they must also breathe out through the nose.

The application named Balloon Game is an interactive game where students practise the type of breathing learned in the Coherence Coach application, but without following the rhythm of the ball. Specifically, in the Balloon Game application, a hot-air balloon appears, flying over different scenarios with variable degrees of speed, depending on the participants’ pace of breathing. If they breathe under parameters similar to those learned in the Coherence Coach application, the journey will be faster, while if they do not breathe under these parameters, the journey will be slower.

### 2.4. Procedure

The programme was implemented in two phases. First, since the intervention was designed for the tutor to train students individually, the pre-intervention phase was designed, where teachers were trained to learn to use the EmWave software’s Coherence Coach and Balloon Game applications [59] so that, later on, they could apply this intervention to students. This training’s goal was two-fold. On one hand, the objective was to teach how to use the programme and the instruments necessary to use it in the classroom in order to take individual HRV measurements throughout the intervention. On the other hand, to help teachers to understand the relationship between steady, calm breathing, and HRV. Teacher training lasted 8 h.

Secondly, for the experimental group, the application phase for the intervention programme consisted of five sessions. One session per week was held individually with each student. Each session lasted approximately 15 min and was conducted in the same physical space (a classroom prepared to this end). The idea is for each girl and boy to conduct their session the same days and times of the week. The different activities proposed were directed by the usual teachers who had been previously trained and were supervised by a member of the research team. Moreover, the HRV scores obtained on each record for each session were collected on a record sheet made to this end. A detailed step by step description of the biofeedback training programme can be obtained by the authors. Table 1, summarizes the intervention protocol both in intervention and control groups.

## 3. Statistical Analysis

In order to observe if HRV scores were significantly different upon completion of training, we conducted the Wilcoxon signed-rank test comparing averages before and after the training sessions. Later, a Student *t*-test analysis was performed to examine if averages between pre-test and post-test low HRV and high HRV by cycles were different. We also analysed the gender differences in HRV by using a variance analysis of HRV results in S1 and S5 based on gender and finally, to examine if the effectiveness of the programme was stable from session to session and taking into account the age of the participants, we used an ANOVA analysis using a MIXED method.

## 4. Results

Once the intervention was completed, the five HRV scores collected on the record sheets were processed and analysed. Below we present the results of the Wilcoxon signed-rank test comparing averages before and after the training sessions. We deemed it essential to observe whether HRV scores were significantly different upon completion of training. In addition, scores from the baseline (S1) were compared with those from session five (S5). It should be mentioned that the criteria used by Kelley and Preacher [60] were followed to interpret the effect size.

In terms of low HRV, for the experimental group the average levels were significantly higher at the baseline (S1; *M* = 62.70) than after the intervention (S5; *M* = 31.80), *z* = −10.26, *p* < 0.05, *r* = −0.53. For medium HRV, the average levels were lower on the baseline session (S1; *M* = 14.50) than after the intervention (*M* = 23.08) *z* = −5.98, *p* < 0.05, *r* =−0.34. Finally, for high HRV, the average levels were lower on the baseline session (S1; *M* = 23.04) than after the intervention (*M* = 79.52), *z* = −12.37, *p* < 0.05, *r* = −0.71. Comparing these same values for the control group, no statistical differences could be observed in any of the three levels of HRV. Low HRV values (S1; *M* = 60.21); (S5; *M* = 62.70), *z* = 0.87, *p* > 0.05, *r* = 0.13. For medium HRV, (S1; *M* = 16.76) and S5 (*M* = 17.09) *z* = 0.73, *p* > 0.05, *r* = 0.12. Finally, for high HRV, (S1; *M* = 24.17) and S5 (*M* = 24.98) *z* = 0.44, *p* > 0.05, *r* = 0.04.

As there were no statistical differences for the control group, we examined differences just for the experimental groups and taking into account the main goal of this investigation, that is to examine if there were differences between age ranges. As follows, with the analysis by age ranges (first, second and third cycle), in Table 2, it can be observed that there is a differential effect of age in relation to high HRV. With changes being statistically significant in all cycles, a lower effect size is observed in the first cycle (7 years old), being a moderate effect size. In the second cycle, a large effect size is observed and in the third cycle, the biggest effect size is observed.

In relation to results by ages of low HRV in Table 3, we also observe the programme’s differential impact. In this case, with the change effect being statistically significant in all cycles, the effect’s lesser size occurs again in the first cycle, this being a moderate effect size. In the second cycle, we observe a large effect size. And in the third cycle, the greatest effect sizes of all cycles occur.

In other words, as the children are older, the greater the impact of the programme (measured through the increased high HRV and reduced low HRV).

In order to give an answer to the third exploratory hypothesis, we analysed the gender differences in HRV. Table 4 shows the results.

None of the HRV scores compared between girls and boys in sessions one and five showed statistically significant results. Thus, there are no differences between both genders in HRV values.

At this point of the study, we also asked if the effectiveness of the programme is stable from session to session across the five sessions and also if there are any differences in each session regarding the age of the participants. In order to answer to these questions, the evolution of high HRV values from session to session was examined by an ANOVA analysis using a MIXED method. The fixed effect estimation showed that the increase of the high HRV observed in each session was statistically significant. See Table 5.

In Figure 1 the graphical evolution of the high HRV of all samples can be seen.

Examining the results, we could observe that the pattern of the first cycle was a little different from the total samples. The fixed effect estimation indicates that there is an increment of the high HRV from S1 to S2, however there a decrease from S2 to S3. This reduction only happens in children of the first cycle, but not in the second and third cycle. See Table 6 and Figure 2.

In the following the graphical evolution of the high HRV of students of cycle 1 can be seen.

## 5. Discussion

As set forth in the study’s hypotheses, it was expected that the breathing-based biofeedback programme would increase HRV. In other words, that the HRV biofeedback intervention programme would be effective if, upon its completion, participants were able to increase their high HRV. It was also expected that such values will be higher for the experimental group compared to the control group. We also intended to explore if there were differences between boys and girls and, most importantly, if the age of the participants had an effect on the results.

Upon analysing the comparison of pre-test (S1) and post-test (S5) high HRV averages from the experimental group, we observe that intervention programme improved high levels of HRV. We also observed that comparing the values of the HRV for the control group, the pre-test and post-test results did show statistically significant differences, showing for this group the same values in both measures. Therefore, we consider that the intervention programme increased the HRV of participants, which can be considered as an indicator of their improving capacity for self-regulation. Emotional self-regulation is a fundamental tool to develop students′ effort, motivation, and personal responsibility about learning and to guarantee scholastic adaptation [14,61,62]. Throughout the primary education period, children experience complex feelings without having internalized the ability to communicate and self-regulate their emotions effectively yet. Therefore, this kind of programme can be essential for children′s well-being by promoting the ability to self-regulate their own emotions, attention and behaviour.

The improvement of the treatment group in HRV observed in our research, matches results observed in other interventions with other age groups. For example, Kuppusamy, et al. [63] carried out an intervention with 520 adolescents (13–18 years) to analyse the impact of a 6-month programme focused on voluntary breathing control on domain parameters of frequency and HRV. After the intervention, it was observed that the experimental group obtained a statistically significant improvement in comparison with the control group in the indicator for HRV’s parasympathetic influence. Mei et al. [9] conducted an intervention with 60 subjects, where they analysed the differences in HRV of participants in two groups treated differently: one group with an HRV biofeedback session, and a group with an autogenous training session. The results revealed significant effects in both interventions on HRV values.

We also found research conducted with the same HeartMath technology used in our research [59]. Field et al. [64] conducted an intervention with 13 participants aged between 26 and 62 years. After the intervention, the experimental group showed greater HRV amplitude than the control group in a statistically significant fashion and with a great effect size (d = 1.97). Aritzeta et al. [45] conducted an HRV biofeedback intervention programme with 152 university students (average age = 19.6), and a control group with 81 university students (average age = 19.4), in order to reduce anxiety levels before exams, and thus improve academic performance. The results indicate a significant improvement in HRV scores, with an effect size of η^2^ = 0.77.

Regarding the gender variable, we observed significant differences between boys and girls in HRV values. The effect size was low in all cases (*d* =< 0.3). In this regard, it should be mentioned that we did not find similar research conducted on the schoolchildren population to examine differences based on gender, which is possibly explained by limited sample sizes. Thus, having observed our results and those shown in the scientific literature [57,58], we cannot conclude that there are evident differences between boys and girls in HRV. Just like Hill et al. [56], we believe that it is necessary to research further into the impact of the gender variable on HRV to observe more conclusive results.

One of the goals of this article was also to observe the effect of the programme depending the age of the participants. We should remember that, given the curricular configuration of primary education, the intervention programme in this research was applied to three different age tranches: cycle one (7 years, n = 76), cycle two (8–9 years, n = 240), and cycle three (10–11 years, n = 47). The results showed that the differences in the pre-test and post-test averages in high HRV were statistically significant in all cycles. However, it is important to note that, in terms of the effect size, there were differences between the cycles: in cycle one (*d* = −0.28), being low to moderate, while in cycle two (*d* = −0.83) and in cycle three (*d* = −0.79), the effect size was large.

In other words, in cycle one, unlike cycle two and cycle three, we observe a moderate size effect both in high HRV and in low HRV. As such, we may suppose that these differences between cycles are associated with shared progression factors inherent to students in cycle one. Differences may be justified by maturity factors related to the age of 7 years. Therefore, specifically heeding to the neuro-anatomical cerebral zones for emotional self-regulation (for example, the anterior ventral cingulate cortex and the prefrontal ventromedial cortex) [65], and being aware of their developmental process between childhood and adolescence [66,67] we can infer that this lower neuro-anatomical maturity progress at 7 years makes it difficult to acquire the emotional self-regulation skill measured in this case through increased high HRV. As such, it should be mentioned that the results observed in our study are also corroborated by those observed in other pieces of research with similar methodologies [64,68,69].

## 6. Conclusions

Based on the results observed in this study, and through the statistical analyses conducted after applying our intervention programme, we can affirm that students who participated in our programme learned to increase their HRV by practising long, steady breathing, at a pace of six pairs of breaths per minute. These results fall in line with other similar studies [23,67], and they are in accordance with the Polyvagal Theory [19,70,71], which affirms that biofeedback procedures based on breathing strategies have an influence on the regulation and improvement of HRV. Moreover, HRV biofeedback has proven to be an effective strategy for children to learn to self-regulate and bolster voluntary breathing learning.

A growing body of psychological research supports the idea that HRV could be an objective physiological measurement to assess emotional responses [22,23,72], including those related to emotional regulation [15,67,73]. High HRV levels correlate to positive results in psychological adjustment in children [74,75,76], adolescents, and adults [27]. It has also been directly linked to self-regulatory capacities in children [77,78,79]. The improvement of emotional self-regulation capacities is very important in Primary Education because at this period there is an increase in academic and social demands that can negatively affect emotional stress [80]. High HRV results are positively associated with emotional self-regulation abilities, which, in turn, positively influences psychological well-being [81] and academic performance [2]. Given that, in childhood, emotional reactivity is highly intense [82] and emotional self-regulation capacity is under development [83,84,85]. Given that childhood is a key period for learning to emotionally self-regulate [86,87,88], it is suitable to use resources to make emotional reactivity regulation possible.

School is an important socialising agent that, along with families, plays a vital role in the promotion of positive mental health in children [89,90]. Fostering healthy relationships during primary education stage is essential to children´s positive experience of school, to promote their well-being and their cognitive and emotional development. For all this, the need to implement programmes based on emotional regulation abilities in academic contexts is justified, and it should also be mentioned that these programmes are especially relevant for children from 8 years old.

### Difficulties, Limitations, and Future Lines of Research

Given the design of the natural groups, it was not possible to maintain equivalent sample proportionality, nor to maintain homogeneous sample sizes throughout cycles. The samples are not representative, and the generalisation of results must be taken with caution.

Regarding collection of HRV data, two kinds of data contamination may have occurred: “the examiner effect”, which refers to the influence of the examiner and of the examiner’s interaction in examining data collection; and “situation effects”, which refer to the influence of different factors on the results of the execution [91]. To attempt to reduce both effects, teachers were selected as “examiners”, and school classrooms were selected as a suitable location for the intervention.

HRV was exclusively analysed in this intervention. It would be advisable to conduct an analysis on the impact of this type of programme on other variables, such as on academic performance. It would also be suitable to analyse the impact the intervention programme had on social relationships. For example, one might establish the presumption that students who participate in the intervention will display more social skills than before participating in the programme, and that participants will also display better social skills than those who do not participate in the programme.

After measuring the efficacy of our intervention, it would be suitable to study the longitudinal effect of the programme and it would be suitable to conduct further HRV biofeedback interventions with similar procedures, samples, and methods, so as to provide greater robustness to the already-proven effectiveness of these interventions.

In closing, we should mention that, although HRV biofeedback interventions are recently and rarely applied in schools, given the high effectiveness of the intervention conducted in this study; and the relevance and need for emotional self-regulation in healthy psychosocial development, we believe that this new field has great potential to boost an emotional self-regulation resource, which is prolonged and steady breathing, in the classroom and in universal fashion, free of cost, for all Primary Education students.

## Figures and Tables

**Figure 1 ijerph-19-05475-f001:**
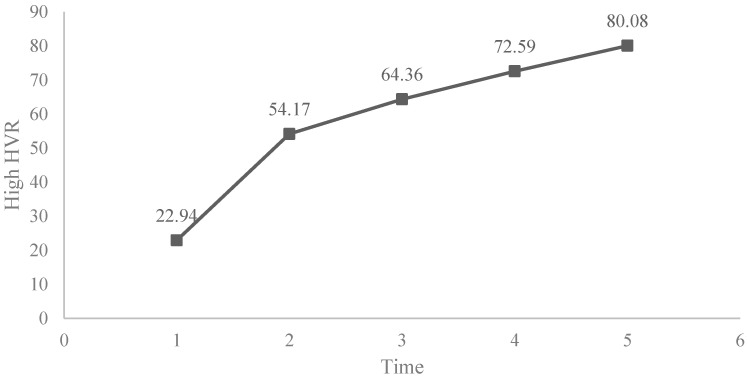
Evolution of the high HRV in all samples.

**Figure 2 ijerph-19-05475-f002:**
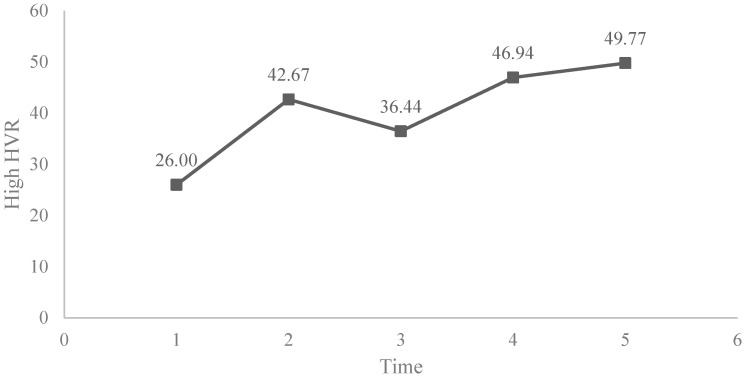
Evolution of the high HRV in cycle 1.

**Table 1 ijerph-19-05475-t001:** Intervention processes for intervention and control groups.

Groups	S1	S2	S3	S4	S5
Intervention	HRV- Base line	Breath Training + HRV-Balloon Game Connexion	Breath Training + HRV-Balloon Game Connexion	HRV-Balloon Game Connexion	HRV-Balloon Game Connexion
Control	HRV- Base line 1	No training	No training	No training	HRV-Base line 2

**Table 2 ijerph-19-05475-t002:** Comparison of S1* and S5* averages by high HRV and cycles.

	Pre-Test			Post-Test		Student’s t	Cohen’s *d*
	N	M	SD	M	SD	*p*	*d*
**Total**	300	23.26	29.56	79.52	60.47	0.000	−1.250
Cycle 1	87	26.61	31.93	49.77	51.10	0.000	−0.558
Cycle 2	175	23.78	29.75	94.99	60.56	0.000	−1.577
Cycle 3	38	13.16	19.88	76.39	54.84	0.000	−1.692

* S1 and S5 numbers refer to sessions 1 and 5.

**Table 3 ijerph-19-05475-t003:** Comparison of S1–S5 averages by low HRV and cycles.

	Pre-Test			Post-Test		Student’s t	Cohen’s *d*
	N	M	SD	M	SD	*p*	*d*
**Total**	299	62.63	30.78	32	30.07	0.000	−1.007
Cycle 1	87	62.16	33.09	31.45	37.23	0.000	−0.506
Cycle 2	174	61.39	31.01	25.32	27.26	0.000	−1.238
Cycle 3	38	69.37	23.18	30.95	29.12	0.000	−1.469

**Table 4 ijerph-19-05475-t004:** Variance analysis in HRV results in S1* and S5* based on gender.

	Average			SD			F		
HRV	N	Girls	N	Boys	Girls	Boys	F	*p*	Cohen’s *d*
Low HRV S1	138	61.80	163	63.21	31.35	30.18	0.16	0.69	−0.046
Low HRV S5	138	33.82	163	30.45	31.26	29.12	0.93	0.34	0.112
Medium HRV S1	138	14.04	163	14.93	9.10	10.13	0.63	0.43	−0.093
Medium HRV S5	138	23.32	163	22.96	23.03	19.74	0.02	0.88	0.017
High HRV S1	138	24.14	163	22.35	30.09	29.10	0.28	0.60	0.061
High HRV S5	138	80.22	163	78.45	63.11	58.01	0.06	0.80	0.029

* S1 and S5 numbers refer to sessions 1 and 5.

**Table 5 ijerph-19-05475-t005:** Evolution of high HRV values from session to session in all samples.

Predictor	Sum of Squares	df	Mean Square	F	*p*
Session 1 high HVR	4616.81	2	2308.41	2.70	0.06
Session 2 high HVR	24578.54	2	12289.27	11.09	0.001
Session 3 high HVR	102411.57	2	51205.78	24.52	0.001
Session 4 high HVR	79778.77	2	39889.38	15.61	0.001
Session 5 high HVR	119275.36	2	59637.69	18.18	0.001

**Table 6 ijerph-19-05475-t006:** Evolution of high HRV values from session to session in cycle 1.

Predictor	Estimation	SE	df	t	*p*	IC95% [LL, UL]
Session 1 high HVR	26.00	4.04	291.64	6.43	0.001	[18.04, 33.95]
Session 2 high HVR	42.66	4.04	291.64	10.56	0.001	[34.71, 50.61]
Session 3 high HVR	36.43	4.04	291.64	9.01	0.001	[28.48, 44.38]
Session 4 high HVR	45.80	4.15	291.64	11.02	0.001	[18.04, 33.95]
Session 5 high HVR	49.77	4.04	291.64	12.31	0.001	[18.04, 33.95]

## Data Availability

Data is available on request from corresponding author.

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
