# Peer review of "Emotional Self-Regulation in Primary Education: A Heart Rate-Variability Biofeedback Intervention Programme"

_ijerph, 2022, doi:10.3390/ijerph19095475_

Round 1

Reviewer 1 Report

The topic is interesting even if it is widely studied. The introduction shows a good knowledge of the literature from the authors. However, some methodological and analytic problems should be solved:

  • Line 125: the sentence is not ended.
  • Are the children in intervention and control groups chosen randomly?
  • The protocol is not clear. Please provide a figure describing the intervention protocol both in intervention and control groups.
  • Lines 175-183: How the intra-beats intervals data are preprocessed? E.g., how are the ectopic beats and missing values treated? Which parameters of HRV (e.g., RMSSD, SDNN, HF, and LF) did you take into consideration? How did you create the 100 points score? Please better describe the HRV analysis and provide an analysis of other HRV parameters (e.g., RMSSD, SDNN, HF, and LF).
  • Please provide the statistical analysis subsection at the end of the method section.
  • Line 218: Why did you use Wilcoxon signed-rank test instead of paired t-test? Did you check for the normality of the data distribution?
  • Line 255: Why did you choose a non-parametric test instead of the parametric one? Probably evaluating the interaction of Two-way ANOVA is the right choice to compare the change in male and female in two-time points.

In conclusion, the main problem of this work is the statistical approach. The statistical analysis is not described in the method section and it is not performed rigorously (no assumption checked and the analytical approach have to be better modelled).

Author Response

Reviewer 1:

    1. Line 125: the sentence is not ended.
  • The sentence is now ended as follows: “Based on previous results, we expect that; compared to a control group, the values of the HRV of the experimental group will be higher than the control group values after the intervention”.
  •  
    1. Are the children in intervention and control groups chosen randomly?
  • Classrooms were randomly assigned to experimental and control groups, however, all students belonging to those classrooms participated in the study. So, children were not randomly assigned to the treatment/non-treatment conditions, but classrooms were. We did mention it in the revised version of the paper as follows:Student participation was voluntary and consented to by the school board, parents, and guardians. Classrooms were randomly assigned to experimental and control groups, however, all children, except those with a clinical diagnosis (i.e., hyperactivity, depression or anxiety disorders) belonging to those classrooms participated in the study. So, children were not randomly assigned to the treatment/non-treatment conditions.
  •  
  •  
    1. The protocol is not clear. Please provide a figure describing the intervention protocol both in intervention and control groups.
  • We have clarified the intervention protocol both in the intervention and the control groups adding a new table. See Table 1.
  •  
    1. Lines 175-183: How the intra-beats intervals data are pre-processed? E.g., how are the ectopic beats and missing values treated?
  • We could not process the intra-beats intervals or the ectopic values, as we could not export raw data of the biofeedback sessions to other software such as Kubios. The main reason was that EmWave based biofeedback coherence ratio values were registered in a pencil-paper format at the school. Due to methodological and computer problems (the school´s computers deleted all sessions every day so, when we were ready to export the data they disappeared) we could not examine the raw scores of each session.  
  •  
    1. Which parameters of HRV (e.g., RMSSD, SDNN, HF, and LF) did you take into consideration?
  • Although there is scientific evidence of using Coherence Ratios offered by the EmWave sofware (e.g., Aritzeta et al., 2017), we only provided data regarding the Coherence Ratios which are based on parameters of HRV.

Aritzeta, A., Soroa, G., Balluerka, N., Muela, A., Gorostiaga, A., Aliri. J. Reducing anxiety and improving academic performance through a biofeedback relaxation training program. Appl. Psychophysiol. Biofeedback 2017, 42, 193-202. https://doi.org/10.1007/s10484-017-9367-z

    1. How did you create the 100 points score? Please better describe the HRV analysis and provide an analysis of other HRV parameters (e.g., RMSSD, SDNN, HF, and LF).
  • Due to the aforementioned research circumstances, we considered, for the sake of parsimony and simplicity, that focusing on the algorithm of Coherence Ratio offered by the EmWave software would be fair enough to contrast the hypothesis (see, for example, other published research which do the same, Aritzeta et al., 2017; Blum et al., 2019; McCraty, 2017). This Coherence Ratio is an algorithm that transforms the HRV into a 0-100 point scale segregated into low, medium, and high values for each depending on the HRV parameters of each participant. The sum of the three values (low, medium, and high) always gives a total of 100 points, making it easier to interpret the output of the each session.

Aritzeta, A., Soroa, G., Balluerka, N., Muela, A., Gorostiaga, A., Aliri. J. Reducing anxiety and improving academic performance through a biofeedback relaxation training program. Appl. Psychophysiol. Biofeedback 2017, 42, 193-202. https://doi.org/10.1007/s10484-017-9367-z

Blum, J., Rockstroh, C., Göritz, A. Heart rate variability biofeedback based on slow-paced breathing with immersive virtual reality nature scenery. Front. Psychol. 2019, 10, 2172. https://doi.org/10.3389/fpsyg.2019.02172

McCraty, R. New frontiers in heart rate variability and social coherence research: techniques, technologies, and implications for improving group dynamics and outcomes. Front. Public Health 2017, 5, 267. https://doi.org/10.3389/fpubh.2017.00267

    1. Please provide the statistical analysis subsection at the end of the method section.
  • The “Statistical Analysis” subsection at the end of the method section has been added and a new paragraph was also included. See below.

“3. Statistical analysis

In order to observe if HRV scores were significantly different upon completion of training, we conducted Wilcoxon signed-rank test comparing averages before and after the training sessions. Later a Student test-test analysis was performed to examine if averages between pretest and postest low HRV and high HRV by cycles were different. We also analysed the gender differences in HRV by using a variance analysis in HRV results in S1 and S5 based on gender and finally, to examine if the effectiveness of the programme was stable from session to session and taking into account the age of the participants, we used an ANOVA analysis using a MIXED method”.

    1. Line 218: Why did you use Wilcoxon signed-rank test instead of paired t-test? Did you check for the normality of the data distribution.
  • We decided to use a Wilcoxon single-rank test in order to combine non-parametric test with parametric ones, in this way we expected to offer more robust results. Moreover, we understood that the Wilcoxon test will also offer a clear and easy to understand image of results. We also understood that these analyses does not jeopardise the test of hypothesis. Nevertheless, a T-test could also be made if the reviewers consider necessary, though we would have probably reached to same conclusions.
  •  
    1. Line 255: Why did you choose a non-parametric test instead of the parametric one? Probably evaluating the interaction of Two-way ANOVA is the right choice to compare the change in male and female in two-time points.
  • The non-parametric test was only used in the first comparison based on the aforementioned reasons. The rest of the analyses were T-tests and variance analysis.
  •  
    1. In conclusion, the main problem of this work is the statistical approach. The statistical analysis is not described in the method section and it is not performed rigorously (no assumption checked and the analytical approach have to be better modelled).
  • Thank you very much for reading our paper and for all your comments that we have tried to address. With the improvements we made in the paper’s reviewed version, we think it has improved its quality.

Reviewer 2 Report

Major comment:

  1. The physiological aspect (justification of the working hypothesis) is weak. Authors presented much data and many references on epidemiology of HRV variability (different age groups). For the scientific justification of HRV modification authors used the Polyvagal Theory. Such a theory indeed exists, though rather in a form of hypothesis, as a part of the Triune Brain theory of Paul McLean (only cal. 120 references in PubMed is found). Such justification is, to my mind, is not fully relevant, though interesting. I would propose to supplement the current justification with papers on the baroreflex sensitivity (BRS) and cardiorespiratory coupling (CRC), as a more traditional physiological ground of the study. Indeed, authors used biofeedback respiratory practices to increase HRV, and concept of BRS and CRC fits well the working hypothesis. Numerous reviews on BRS and CRC are available from PubMed.

Altogether, I would propose to modify the Introduction section in the direction of physiological mechanisms, rather than neuromorphological or behavioral.

Minor comments:

Introduction, line 81. A debatable statement. HRV is not necessarily dependent on HR. In some instances, HRV is low at low HR, and high at high HR.

Introduction, line 125. Sentence is not finished?

Material and Methods, line 143. What is the purpose to present the average age of the entire group of subjects (8,51 years)? Actually three DIFFERENT age groups were studied, and I would like to know the age characteristic of these 3 particular groups.

line 146. What is Dt? Should it be St, SD or ME (standard deviation or mean error)?

line 150. I think that the term "cycle" is not relevant to define the study group. The term "cycle" prompts smth linked to a process, rather than a group. Should it be "age group I" or suchlike?

line 166-169. S1 and S5 are acronyms for "session", not for a measurement? Please, write "The first session (S1)..." or "Measurements during the first session (S1)..."

Also, where is description for S2-4? They should stay close to conceive s design. Only far further in the text S2-4 are found.

line 178. What is "coherence level"? Coherence of what with what? Respiratory cycle to HR?

Also, which HRV parameters are used in the software to compute HRV "levels"? There time-, frequency-domain and nonlinear parameters (about 20-30, altogether).

Results

line 294 (Figure 2). Why not to present data also on the age groups 2 and 3 (cycles 2 and 3) to better conceive peculiarities of the result for all three groups?

Discussion

line 307. What is time 1 and time 2. Better to explain it with words, as it is difficult to follow the idea.

How the Polyvagal Theory explains the outcome?

The comment is same, as for the Introduction section: please discuss the outcome in a "more physiological" aspect.

Conclusion

Results are solid, but they need better physiological interpreting.

Author Response

Reviewer 2

2.1.) The physiological aspect (justification of the working hypothesis) is weak. Authors presented much data and many references on epidemiology of HRV variability (different age groups). For the scientific justification of HRV modification authors used the Polyvagal Theory. Such a theory indeed exists, though rather in a form of hypothesis, as a part of the Triune Brain theory of Paul McLean (only cal. 120 references in PubMed is found). Such justification is, to my mind, is not fully relevant, though interesting. I would propose to supplement the current justification with papers on the baroreflex sensitivity (BRS) and cardiorespiratory coupling (CRC), as a more traditional physiological ground of the study. Indeed, authors used biofeedback respiratory practices to increase HRV, and concept of BRS and CRC fits well the working hypothesis. Numerous reviews on BRS and CRC are available from PubMed.

  • Thank you very much for your suggestion. To our knowledge the Triune Brain Theory is unknown, and agreeing that it could reinforce the theoretical basis of this article, we consider that it would also reduce the parsimony of the hypothesis development. We think that the Polyvagal Theory is a robust ground with hundreds of scientific publications that offer us solid bases for hypothesis development.

Minor comments:

2.2) Introduction, line 81. A debatable statement. HRV is not necessarily dependent on HR. In some instances, HRV is low at low HR, and high at high HR.

  • We do agree, however, we understand that the sentence does not directly relate HRV with HR, but how the suppression of the vagal brake increases the HR pace.

2.3) Introduction, line 125. Sentence is not finished?

  • Absolutely. Thank you! The sentence is now ended as follows: “Based on previous results, we expect that; compared to a control group, the values of the HRV of the experimental group will be higher than the control group values after the intervention”.
  •  

2.4) Material and Methods, line 143. What is the purpose to present the average age of the entire group of subjects (8,51 years)? Actually three DIFFERENT age groups were studied, and I would like to know the age characteristic of these 3 particular groups.

  • Additional demographic data has been added. See below.

The experimental and control groups were divided into the three cycles: 29.2% belonged to the first cycle (2nd grade, M = 7.58 years and SD = 0.38), 57.7% to the second cycle (3rd and 4th grades, M = 8.93 years old and SD = 0.85), and the remaining 13.1% were third-cycle students (5th and 6th grades, M = 11.04 years old and SD = 0.91).

2.5) line 146. What is Dt? Should it be St, SD or ME (standard deviation or mean error)?

                à Thank you. Yes, we have changed the abbreviations and we have removed the commas and added dots in tables. We changed de abbreviations and errors found in the text. DT AND ST were SD. And Mdn was M.

2.6) line 150. I think that the term "cycle" is not relevant to define the study group. The term "cycle" prompts smth linked to a process, rather than a group. Should it be "age group I" or suchlike?

à Yes, somehow it could be a bit misleading, however, instead of changing the term cycle in all the article, we have clarified it the first time it was used. See the paragraph below.

The experimental and control groups were divided into the three cycles: 29.2% belonged to the first cycle (2nd grade, M = 7.58 years and SD = 0.38), 57.7% to the second cycle (3rd and 4th grades, M = 8.93 years old and SD = 0.85), and the remaining 13.1% were third-cycle students (5th and 6th grades, M = 11.04 years old and SD = 0.91).

2.7) line 166-169. S1 and S5 are acronyms for "session", not for a measurement? Please, write "The first session (S1)..." or "Measurements during the first session (S1)..."

à Absolutely. We have rewritten the paragraph as follows:

The first measurement of session 1 (S1) is the pre-treatment measurement, and informs about the baseline HRV numbers. The 5th measurement of session 5 (S5) is the post-treatment measurement. For the control group only two measures were taken in session 1 (S1) and session 5 (S5) and at the same period of time than the experimental group.

2.8) Also, where is description for S2-4? They should stay close to conceive s design. Only far further in the text S2-4 are found.

à In order to clarify the design and procedure, a new table has been added. See Table 1 in the reviewed manuscript. (Suggested by Reviewer 3)

Table 1, summarizes the intervention protocol both in intervention and control groups.

Table 1. Intervention processes for intervention and control groups.

Groups

S1

S2

S3

S4

S5

Intervention

HRV-Base line

Breath Training + HRV-Balloon Game Connexion

Breath Training + HRV-Balloon Game Connexion

HRV-Balloon Game Connexion

HRV-Balloon Game Connexion

Control

HRV-Base line 1

No training

No training

No training

HRV-Base line 2

2.9) line 178. What is "coherence level"? Coherence of what with what? Respiratory cycle to HR?

à We have changed the “level” concept to the “ratio” concept. And also have clarified the meaning of low, medium, and high coherence ratios. See below.

Using a mathematical algorithm, the EmWave software transforms the collected data (e.g. RMSSD, SDNN...) into low, medium, and high coherence ratios. These low, medium and high coherence ratios are directly related to HRV amplitude rates. Thus, a low coherence level matches low HRV, a medium coherence level matches medium HRV, and a high coherence level matches high HRV.

2.10) Also, which HRV parameters are used in the software to compute HRV "levels"? There time-, frequency-domain and nonlinear parameters (about 20-30, altogether).

à The algorithm used by the EmWave software is not public. However, it has been used in other publications as an indicator of HRV ratios (Aritzeta et al., 2017; Blum et al., 2019; McCraty, 2017). We do agree that a more detailed analysis using, for example, the Kubios HRV software and examining the time and frequency domains indicators would have been much more precise, however, unfortunately, due to design problems; we could not use these data in our study.

Results

2.11) line 294 (Figure 2). Why not to present data also on the age groups 2 and 3 (cycles 2 and 3) to better conceive peculiarities of the result for all three groups?

à With respect to the evolution patterns of cycles 2 (age group II) and 3 (age group III), they were basically the same and no remarkable differences could be observed compared to the full sample.

Discussion

2.12) line 307. What is time 1 and time 2. Better to explain it with words, as it is difficult to follow the idea.

à We have changed the sentence as follows.

We also observe that comparing the values of the HRV for the control group, the pretest and postest results did show statistically significant differences, showing this group, the same values in both measures.

2.13) How the Polyvagal Theory explains the outcome?

As this theory addresses the interrelation between the vagus nerve and the emotional experience, and Primary Education children completely possess the vagus nerve system and the ventral-vagus branch and the dorsal-vagus branch of the vagus nerve, provide an inhibitory entry to the heart through the parasympathetic nervous system (reducing heart rate) and influencing the HRV outcomes, when children receive the intervention the environment is perceived to be safe, thus, the ventral vagal complex is activated, leading to an increased influence of the myelinated vagal channels that slow down heartbeat frequency, increasing HRV, and inhibiting the fight-or-flight mechanisms of the sympathetic nervous system.

2.14) The comment is same, as for the Introduction section: please discuss the outcome in a "more physiological" aspect.

à Thank you again for your suggestion. We really appreciate it. Following the suggestion of other reviewers, we have made an effort to better connect the Polyvagal Theory to our results in terms of emotional self-regulation and HRV. We understand that a “more physiological” discussion of our results will also need a restructuring of the article theoretical framework. We understand that the goal and hypothesis of this paper are well justified by the Polyvagal Theory.

Conclusion

2.15) Results are solid, but they need better physiological interpreting.

à Thank you for your suggestion. We refer to answers given in suggestion 2.1 and 2.13.

Reviewer 3 Report

This study conducted a bio-feedback intervention on children aged 7 to 11 years old to train them in emotional self-regulation (operationalized as HRV). Given the increase in the prevalence of mental illness among children in many countries, the evidence-based intervention programs targeted at this age population is especially valuable. The use of biofeedback techniques in the intervention is also a highlight. I would recommend the paper for publication after several issues be addressed.

  1. The introduction section could be tightened up a little to focus more on the importance of emotional self-regulation for children and the HRV as a useful index thereof. Biofeedback technique could enhance the effectiveness of the training, but unless the authors intended to compare the effectiveness of different training technique (intervention with or without biofeedback), it would be misleading and detrimental to the logical flow of the paper to start the paper by mentioning the advantages of biofeedback.
  2. The introduction of Polyvagal theory, and its relations with HRV and biofeedback, could be enhanced. In its current form, the relevant passages were lengthy and ponderous.
  3. The use of some numerical notation and abbreviations are uncommon. For example, in Line 144 to 146, what is Dt? In Line 232, what is Mdn? In the Tables, what are the “St”? Are they SD (standard deviation)? Also in the tables, it seems that decimal commas were used instead of dots, yet the authors used decimal dots in the main text.
  4. The psychological/physiological significance of the low, intermediate, and high HRV is missing in the method section. The use of three types of HRV indices is confusing without a proper explanation, especially because the use of “high HRV” and “low HRV” in Lines 74-85 seems to suggest that HRV is a single index.
  5. It is appropriate to adopt a mixed design ANOVA (or split-plot design) to analyze data from this research, but the author did not report the details of this ANOVA, except in Table 4, which lacks a lot of information. Did they conflate the three cycles and only analyze data using a (5 sessions * experimental conditions) design, or the three cycles were retained and were treated as nested? The authors should shift the focus of the Result section to this mixed design ANOVA, as their main research question could well be answered by a thorough analysis of the mixed ANOVA alone.
  6. In the discussion section, the theoretical and practical significance of children’s emotional self-regulation were only superficially mentioned, which weakened the value of this study. The use of HRV is only one small fraction of that significance, i.e., a relatively novel and convenient method; but for study implication, it is the emotional self-regulation that is really the focal point.

Author Response

Reviewer 3:

We would like to especially thank the suggestions made by Reviewer 3. It really helped to improve the research article.

3.1) The introduction section could be tightened up a little to focus more on the importance of emotional self-regulation for children and the HRV as a useful index thereof.

                à We have included additional sentences that should tighten up the association between the self-regulation process for children and the HRV. Below you can read the paragraph we have added:

“In childhood, emotions are experienced with great intensity and with low capacity for emotional regulation (Silvers et al., 2017) which may have a negative impact on psychosocial skills, school performance and well-being (Datu & King, 2018; Golding et al., 2019). During primary education, the capacity for emotional self-regulation is still in development which offers an opportunity to implement interventions with the goal of improving emotional self-regulation abilities (Thomas et al., 2017). Specifically, interventions based on biofeedback of the HRV and focused on breathing have been shown to be effective in improving emotional self-regulation capacity in children (Cruz, 2019)”.

Cruz, A. (2019). Biofeedback as an intervention to increase self-regulation in school-aged children in an urban charter school. Tesis doctoral presentada en la Facultad de School of Human Service Professions. Pensilvania U.S.A: Widener University. https://search.proquest.com/docview/2284756111/fulltextPDF/20682CE768684BC1PQ/1?accountid=17248

Datu, J. A. D., King, R. B. (2018). Subjective well-being is reciprocally associated with academic engagement: a two-wave longitudinal study. Journal of School Psychology69, 100–110. https://doi.org/10.1016/j.jsp.2018.05.007

Goldin, P. R., Moodie, C. A., Gross, J. J. (2019). Acceptance versus reappraisal: behavioral, autonomic, and neural effects. Cognitive, Affective, y Behavioral Neuroscience19(4), 927–944. https://doi.org/10.3758/s13415-019-00690-7:

Silvers, J.S., Insel, C., Powers, A., Franz, P., Helion, C., Martin, R., Weber, J, Mischel, W., Casey, B.J., Ochsner, J.N. (2017). The transition from childhood to adolescence is marked by a general decrease in amygdala reactivity and an affect-specific ventral-to-dorsal shift in medial prefrontal recruitment. Developmental Cognitive Neuroscience, 25, 128-137. https://doi.org/10.1016/j.dcn.2016.06.005

Thomas, J. C., Letourneau, N., Campbell, T. S., Tomfohr-Madsen, L., Giesbrecht, G. F. (2017). Developmental origins of infant emotion regulation: Mediation by temperamental negativity and moderation by maternal sensitivity. Developmental Psychology, 53(4), 611. https://doi.org/10.1037/dev0000279

3.2) Biofeedback technique could enhance the effectiveness of the training, but unless the authors intended to compare the effectiveness of different training technique (intervention with or without biofeedback), it would be misleading and detrimental to the logical flow of the paper to start the paper by mentioning the advantages of biofeedback.

                à We do agree. We have included a new starting paragraph which we understand makes the paper clearer. Please see the paragraph added in the 3.2 suggestion.

3.3) The introduction of Polyvagal Theory, and its relations with HRV and biofeedback, could be enhanced. In its current form, the relevant passages were lengthy and ponderous.

                à We again, do agree with the reviewer. We have restructured and simplified this section trying to make it clearer and enhancing the relationship between Polyvagal theory and its relation with HRV and biofeedback. Below you can read the new structure of the paragraphs.

“This theory addresses the interrelation between the vagus nerve and the emotional experience allowing us to associate self-regulation abilities with values of HRV. The vagus nerve is the tenth cranial nerve and is the main nerve in the parasympathetic division determining emotional self-regulation [10]. Primary Education children completely possess the vagus nerve system [11,12] and the ventral-vagus branch and the dorsal-vagus branch of the vagus nerve, provide an inhibitory entry to the heart through the parasympathetic nervous system (reducing heart rate) and influencing the HRV outcomes.

The vagus nerve system is hierarchically structured into three sequential functional sub-systems: a) The ventral vagal complex or myelinated vagus, b) The sympathoadrenal system, and c) The dorsal vagal complex or non-myelinated vagus [14,15]. When the environment is perceived to be safe, the ventral vagal complex is activated, leading to an increased influence of the myelinated vagal channels that slow down heartbeat frequency, increase HRV, and inhibit fight-or-flight mechanisms of the sympathetic nervous system. This physiological state makes it possible for the prefrontal cortex's structures to work properly, which are in charge of attention and self-regulation, both of which are differentially developed for a child of 7 years old and one of 11 years old. However, and despite Primary Education children completely possess the vagus nerve system, we still do not have enough evidences to know how effective a HRV and breathing based biofeedback programme can be depending of the age of the children”

3.4) The use of some numerical notation and abbreviations are uncommon. For example, in Line 144 to 146, what is Dt? In Line 232, what is Mdn? In the Tables, what are the “St”? Are they SD (standard deviation)? Also in the tables, it seems that decimal commas were used instead of dots, yet the authors used decimal dots in the main text.

                à Thank you. Yes, we have changed the abbreviations and we have removed the commas and add dots in tables. We chanced de abbreviations and errors found in the text. DT AND ST were SD. And Mdn was M.

3.5) The psychological/physiological significance of the low, intermediate, and high HRV is missing in the method section. The use of three types of HRV indices is confusing without a proper explanation, especially because the use of “high HRV” and “low HRV” in Lines 74-85 seems to suggest that HRV is a single index.

               à We have added an additional explanation in the method section about the meaning of high, medium, and low HRV values. Please, read below the sentences added in the revised paper.

Using a mathematical algorithm, the EmWave software transforms the collected data (i.e. RMSSD, SDNN...) into low, medium, and high coherence levels. These low, medium and high coherence levels are directly related with HRV amplitude rates. If HRV rates are low and the power spectrum of it is short, then the low coherence ratios will show higher punctuations. On the contrary if HRV rates are high, then coherence ratios will also show values. Thus, a low coherence level matches low HRV, a medium coherence level matches medium HRV, and a high coherence level matches high HRV.

3.6) It is appropriate to adopt a mixed design ANOVA (or split-plot design) to analyze data from this research, but the author did not report the details of this ANOVA, except in Table 4, which lacks a lot of information. Did they conflate the three cycles and only analyze data using a (5 sessions * experimental conditions) design, or the three cycles were retained and were treated as nested? The authors should shift the focus of the Result section to this mixed design ANOVA, as their main research question could well be answered by a thorough analysis of the mixed ANOVA alone.

                à Thank you very much for your suggestion. We understand that, being true that using a mixed ANOVA could be enough to address the main research question of this investigation, some variables, e.g. gender did not fulfil the standard parameters of normal distribution and that is why we decided to use non parametric test. Regarding the lack of information, we have followed actual APA standards to plot the tables and provide the necessary information to contrast the hypothesis. We hope will be enough.

3.7) In the discussion section, the theoretical and practical significance of children’s emotional self-regulation were only superficially mentioned, which weakened the value of this study. The use of HRV is only one small fraction of that significance, i.e., a relatively novel and convenient method; but for study implication, it is the emotional self-regulation that is really the focal point.

                à Following your advice, we have strengthened the theoretical and practical significance of children’s emotional self-regulation by including additional information in the discussion section. See below the new paragraph added.

High HRV levels correlate to positive results in psychological adjustment in children [72-74], adolescents, and adults [24]. It has also been directly linked to self-regulatory capacities in children [75-77]. The improvement of emotional self-regulation capacities is very important in Primary Education because at this period there is an increase in academic and social demands that can negatively affect emotional stress (Pirskanem et al., 2019). High HRV results are positively associated with emotional self-regulation abilities, which, in turn, positively influences psychological well-being (Kim et al., 2018) and academic performance (Datu & King, 2018).

Datu, J. A. D., King, R. B. (2018). Subjective well-being is reciprocally associated with academic engagement: a two-wave longitudinal study. Journal of School Psychology69, 100–110. https://doi.org/10.1016/j.jsp.2018.05.007

Pirskanen, H., Jokinen, K., Karhinen-Soppi, A., Notko, M., Lämsä, T., Otani, M., Rogero-García, J. (2019). Children’s emotions in educational settings: Teacher perceptions from Australia, China, Finland, Japan and Spain. Early Childhood Education Journal47(4), 417-426. https://doi.org/10.1007/s10643-019-00944-6

Kim, D. H., Bassett, S. M., Takahashi, L., Voisin, D. R. (2018). What does selfesteem have to do with behavioral health among low-income youth in Chicago? J. Youth Studi. 21, 999–1010. doi:10.1080/13676261.2018.1441982

Round 2

Reviewer 1 Report

The authors have partially answered all of my doubts. Two other main problems should be fixed before acceptance.

  1. Did you check if the groups show similar values in the baseline (both anthropometric and HRV parameters)? If not, you should take it into consideration during the analysis (for example by covaring the analysis by baseline values).
  2. The computation method for coherence level is not clear. "Using a mathematical algorithm" is not enough to understand how this value is calculated. Please provide more details and, if possible, the formula to compute it. Moreover, please provide all the metrics used to compute the coherence level.

Author Response

Reviewer 1

Thank you for your second review. Herewith we answer to your suggestions.

  • Did you check if the groups show similar values in the baseline (both anthropometric and HRV parameters)? If not, you should take it into consideration during the analysis (for example by covaring the analysis by baseline values).
  • Yes, we did and there were no statistical differences in the baseline values for all groups.
  • The computation method for coherence level is not clear. "Using a mathematical algorithm" is not enough to understand how this value is calculated. Please provide more details and, if possible, the formula to compute it. Moreover, please provide all the metrics used to compute the coherence level.
  • We agree that using a mathematical algorithm is not enough to understand how the coherence ratios are calculated, however, as we mentioned in our previous answer this algorithm is property of the HeartMath Company and the Emwave software does not provide it. Unfortunately, we cannot provide the formula. Moreover, as we mentioned in our previous answer, we could not process the intra-beats intervals or the ectopic values, as we could not export raw data of the biofeedback sessions to other software such as Kubios. The main reason was that EmWave based biofeedback coherence ratio values were registered in a pencil-paper format at the school. Due to methodological and computer problems (the school´s computers deleted all sessions every day so, when we were ready to export the data they disappeared) we could not examine the raw scores of each session. For this reason, we had to focus on the algorithm of Coherence Ratio offered by the EmWave software to contrast our hypothesis (see, for example, other published research which do the same, Aritzeta et al., 2017; Blum et al., 2019; McCraty, 2017). We hope this answer will be enough.

Reviewer 3 Report

I wish to thank the authors for their thorough revision. The logical flow of the manuscript has been improved significantly. However, the issue regarding ANOVA was essentially unfixed. For example, "Did they conflate the three cycles and only analyze data using a (5 sessions * experimental conditions) design, or the three cycles were retained and were treated as nested?", this seems unfixed. The report of ANOVA results were only partial, for example, only numerator degrees of freedom was shown, but the denominator dfs were not reported.  

Minor

Lines 186-187 and Line 189-190 seem redundant.

Author Response

I wish to thank the authors for their thorough revision. The logical flow of the manuscript has been improved significantly. However, the issue regarding ANOVA was essentially unfixed. For example,

3.1      . "Did they conflate the three cycles and only analyze data using a (5 sessions * experimental conditions) design, or the three cycles were retained and were treated as nested?", this seems unfixed.

  • In table 4, we did conflate the three cycles and, as you mentioned, we analyze data using a 5 session * experimental condition for High HRV values only. In Table 5, we basically did the same but, instead of conflating the three cycles we conduct these analyses just for Cycle 1 (Group I).

3.2.     The report of ANOVA results were only partial, for example, only numerator degrees of freedom was shown, but the denominator dfs were not reported.

  • Thank you for your suggestion. We followed American Psychological Association 7.0 standards for reporting Anova result tables

3.3.     Lines 186-187 and Line 189-190 seem redundant.

  • Indeed. Thank you! We have erased the repetition in 189-190 lines.